# Antioxidants in Down Syndrome: From Preclinical Studies to Clinical Trials

**DOI:** 10.3390/antiox9080692

**Published:** 2020-08-03

**Authors:** Noemí Rueda Revilla, Carmen Martínez-Cué

**Affiliations:** Department of Physiology and Pharmacology, Faculty of Medicine, University of Cantabria, 39011 Santander, Spain; martinec@unican.es

**Keywords:** Down syndrome, antioxidants, oxidative stress, mitochondrial dysfunction, Ts65Dn

## Abstract

There is currently no effective pharmacological therapy to improve the cognitive dysfunction of individuals with Down syndrome (DS). Due to the overexpression of several chromosome 21 genes, cellular and systemic oxidative stress (OS) is one of the most important neuropathological processes that contributes to the cognitive deficits and multiple neuronal alterations in DS. In this condition, OS is an early event that negatively affects brain development, which is also aggravated in later life stages, contributing to neurodegeneration, accelerated aging, and the development of Alzheimer’s disease neuropathology. Thus, therapeutic interventions that reduce OS have been proposed as a promising strategy to avoid neurodegeneration and to improve cognition in DS patients. Several antioxidant molecules have been proven to be effective in preclinical studies; however, clinical trials have failed to show evidence of the efficacy of different antioxidants to improve cognitive deficits in individuals with DS. In this review we summarize preclinical studies of cell cultures and mouse models, as well as clinical studies in which the effect of therapies which reduce oxidative stress and mitochondrial alterations on the cognitive dysfunction associated with DS have been assessed.

## 1. Oxidative Stress (OS) in Down Syndrome (DS)

### 1.1. Mechanisms of OS Involved in DS

Down syndrome (DS), the most common genetic cause of intellectual disability [1], is caused by a partial or complete triplication of the human chromosome 21 (Hsa21). Most of the altered phenotypes of DS arise because of the altered expression of Hsa21 genes [2].

Cognitive dysfunction in DS is due to defects in the growth and differentiation of the central nervous system that appear during early prenatal stages [3,4,5,6,7,8,9]. However, later in life, the cognitive alterations in DS individuals are aggravated because their brains undergo premature aging and present the early appearance of Alzheimer’s disease (AD) neuropathology, which is characterized by amyloid plaque deposits, neurofibrillary tangles (NFTs) caused by hyperphosphorylation of the tau protein, neurodegeneration, synapse loss, and neuroinflammation due to microglial activation which increases the release of pro-inflammatory cytokines [4,10,11,12,13,14]. Furthermore, similar to AD, basal forebrain cholinergic neurons (BFCNs) and noradrenergic neurons progressively degenerate in the DS brain [15,16,17].

Oxidative stress (OS) is one of the most important neuropathological processes responsible for the cognitive alterations and the deficits in neuronal function in DS. Brain tissue can be more susceptible to undergoing elevated levels of OS than other tissues because it is rich in fatty acids which are ideal biomolecules for peroxidation processes, and because it contains low concentrations of antioxidant enzymes and is also characterized by a high aerobic metabolic rate [18]. In DS, as in other neurodegenerative diseases, increased OS is produced both by a high rate of reactive oxygen species (ROS) production and low level of antioxidant enzymes and reducing agents [18,19]. In this syndrome, the increase in OS is produced by the overexpression of several Hsa21 genes [4,13,20] that encode particular proteins which directly or indirectly induce the production of ROS, damaging molecules which are crucial for proper functioning of the cell. One of these Hsa21 genes is *SOD1*, which is responsible for the expression of the enzyme Superoxide Dismutase (SOD1), which transforms superoxide anions into molecular oxygen and hydrogen peroxide (H_2_O_2_). In DS, the increased activity of SOD1 results in the formation of excessive levels of H_2_O_2_ which are not adequately compensated by the activity of two other antioxidant enzymes—catalase (CAT) and glutathione peroxidase (GPx)—creating a redox imbalance. In fact, all tissues from DS patients display an altered SOD1/GPx ratio [21].

Mitochondria are the major site of production of ROS through the oxidative phosphorylation pathway and are one of the main targets of free radicals. In DS, it has been demonstrated that beginning at early developmental stages, there are important mitochondrial structural and functional dysfunctions associated with high ROS production [22,23] due to the excessive generation of O_2_^−^ and H_2_O_2_. The alterations in mitochondrial function found in trisomic cells have been demonstrated to be caused by a loss of equilibrium between mitochondrial biogenesis and turnover [24,25,26]. Furthermore, DS fibroblasts and neurons show damaged mitochondria with anomalies on their cristae [23,25,27,28]. Postmortem brain tissue from DS patients also displays severe alterations in important complex II and V mitochondrial proteins of the mitochondrial chain [29], as well as mitochondrial DNA mutations [30]. Deficits in complex I activity associated with increased ROS production in human DS fibroblasts have also been found [31]. Thus, the mitochondrial dysfunction raises the intracellular ROS production, but also alters the energy metabolism, leading to a lower ATP production [32,33]. These effects play an important role in neuronal dysfunction, affecting synaptic transmission and, as a result, impairing cognitive function [13].

Despite insufficient CAT and GPx activity, in DS the high levels of H_2_O_2_ that are accumulated in the cytosol are not adequately eliminated by other antioxidant enzymes, including glutathione transferase and thioredoxin peroxidase [34]. Thus, the accumulation of H_2_O_2_ leads to the formation of a most harmful hydroxyl radical (^●^OH), which damages cells, mitochondrial membranes, proteins, and other biomolecules. In addition, the DS fetal brain shows decreased expression of peroxiredoxin 2, an antioxidant enzyme involved in lipid and protein protection against oxidative injury [35].

To counteract excessive ROS production, cells also possess other antioxidant compounds including glutathione, amino acids (arginine, taurine, creatine), metals (selenium and zinc), and vitamins (vitamins E and C) [18]. DS individuals also present altered levels of these antioxidant compounds. The glutathione system is affected in DS [36,37]. The levels of glutathione are decreased in these individuals [38,39,40] and increased oxidized glutathione/glutathione ratio has been found in fibroblasts from DS fetuses [36]. Additionally, lower blood and/or plasma levels of other antioxidant molecules such as vitamin E, vitamin C, selenium, and zinc have been found in children and adult DS individuals [38,41,42,43]. Furthermore, fetal DS brains show reduced levels of taurine in the frontal cortex [44]. These alterations in other components of the endogenous antioxidant system may be additional factors which contribute to the exacerbation of oxidative damage in DS.

Another Hsa21 gene, *APP* (Amyloid Precursor Protein), has also been implicated in the increased ROS production characteristic of DS. It has been demonstrated that overexpression of this gene increases the production of the APP protein and leads to the accumulation of β-amyloid (Aβ) peptides, aggravating the redox imbalance [20,45]. Aβ accumulation has been associated with DS-related OS and mitochondrial dysfunction [46,47,48,49]. Accumulation of Aβ in the mitochondria interferes with the respiratory chain and facilitates the formation of membrane permeability transition pores, impairing energy metabolism [18]. In addition, independently of Aβ deposition, APP may cause mitochondrial dysfunction, and this may be an additional source of the increased OS in DS [50].

Besides *APP* and *SOD*, the triplication of other Hsa21 genes such as carbonyl reductase (*CBR*), *BACH1*, and S100β has also been demonstrated to play an important role in the increased OS found in DS individuals [20]. The *BACH1* gene is a transcriptional repressor of specific genes involved in the cell response such as HO-1 (Heme oxigenase-1), which participates in the heme degradation. This enzymatic step produces biliverdin, which is converted to bilirubin by the enzyme biliverdin reductase. Bilirubin, at low concentrations, may act as a physiological antioxidant [51,52]. In DS brains, increased BACH1 protein levels coupled with the reduced induction of HO-1 seem to be involved in the early increase of OS [52].

The *CBR* gene codifies the enzyme carbonyl reductase which detoxifies the cytotoxic metabolic intermediates carbonyls by its reduction. The levels of this enzyme are elevated in different areas of the DS brain due to the enzyme induction produced by elevated carbonyls compounds [53].

The *S100β* gene, which is also triplicated in DS, is responsible for the release of proinflammatory mediators from microglial cells, causing negative effects in neurons. Aberrant S100β production in the brain of DS subjects promotes β-amyloid plaque formation [54]. Furthermore, the overexpression of S100β in human DS neural progenitors increases ROS formation [55]. In rodent macrophages, the S100β protein stimulates the production of nitric oxide (NO) [56]. NO is one of the most common cellular free radicals and when its concentration reaches dangerously high values it undergoes redox actions and produces reactive nitrogen species [18].

### 1.2. Role of Oxidative Stress in DS Neurodegeneration and Cognitive Dysfunction

In addition to affecting the integrity of important cellular components which alters proper neurophysiological processes, OS also alters multiple pathways implicated in cell growth, gene expression, neurodegeneration, and protein function, which plays an important role in DS cognitive dysfunction.

Increased ROS production interacts with and induces damage to proteins, lipids, and DNA, which alters neuronal physiology and function [20,57]. Several studies have demonstrated that neurons from DS individuals present increased levels of different markers of lipid peroxidation, of oxidized proteins, and of DNA damage [58,59,60]. Oxidative damage to proteins modifies the activity of essential receptors, hormones, and enzymes [37], and also affects cellular functions, leading to alterations of the intracellular degradative machinery (i.e., the proteasome system, autophagy, and exosome cargo and release). If these oxidized proteins become dysfunctional, this exacerbates their accumulation, aggravating neurodegeneration in DS individuals [50,61,62]. Oxidative modification of lipids also produces structural and functional damage in cells and mitochondrial membranes [57]. Finally, there is also evidence that suggests that in fetal and adult DS cells, an increase in OS produces an accumulation of DNA damage and defects in DNA repair mechanisms [63,64,65] that leads to genomic instability. In neurons, these alterations play a central role in the degenerative process associated with oxidative damage in DS and other neurodegenerative diseases [66].

In addition, both OS and mitochondrial dysfunction induce premature cell senescence [66,67,68], a process that is characterized by a permanent arrest of cell proliferation [69]. Amniocytes and placentas from trisomy 21 pregnancies present a higher prevalence of senescent cells [70]. In trisomic fibroblasts of human DS fetuses, the increased ROS production together with the subsequent amount of oxidized proteins and the decline in the levels of ATP leads to the acquisition of premature senescent phenotypes [68]. Fibroblasts from children with DS have an increased ratio of SOD1/GPx activity, resulting in high levels of H_2_O_2_ which induces features of cellular senescence [71]. Thus, from prenatal stages, intracellular OS, the decrease in the antioxidant defense system, and mitochondrial dysfunction induce cellular senescence in DS, and this phenomenon is aggravated with aging and the development of AD neuropathology, further exacerbating cognitive dysfunction.

Moreover, several studies have demonstrated that increased levels of ROS in DS neurons alter the processing of the APP protein, inducing the accumulation of Aβ peptides and producing neuroinflammation. Thus, increased OS exacerbates AD neuropathology in DS [11,12,20,46,72].

The elevation of OS in the DS brain from prenatal stages alters proper brain development because it interferes with crucial processes such as neurogenesis, neural differentiation, migration, connectivity, and neuronal survival [22,50]. In later stages of life, OS is exacerbated, contributing to neuropathological changes and degeneration, and to the development and progression of AD neuropathology and to the cognitive dysfunction associated with DS [20,36].

## 2. Brain Oxidative and Mitochondrial Profile in Mouse Models of DS

To study the mechanisms implicated in the neurobiological and cognitive alterations found in DS and to develop therapeutic approaches to reduce or prevent these impairments, several mouse models of DS have been generated. These animals are trisomic for different sets of Hsa21 orthologous genes localized in syntenic regions of three murine chromosomes Mmu16, Mmu17, and Mmu10, and recapitulate most of the neuroanatomical, neurochemical, and behavioral DS phenotypes [2,73,74,75,76]. These murine models present a similar brain oxidative and mitochondrial dysfunction profile to the one previously described for the DS population.

The Ts65Dn mouse, the best characterized and the most commonly used model of DS, shows increased OS (lipid peroxidation and protein carbonylation) and mitochondrial dysfunction in the hippocampus and cortex [77,78,79,80,81,82,83], that affects brain structure and function [75]. Although, to the best of our knowledge, oxidative DNA damage has not been studied in the brain of this model, it has been demonstrated that other cell types such as satellite cells of skeletal myofibers and hematopoietic stem cells accumulate oxidative DNA damage and prematurely develop a senescent phenotype in the Ts65Dn mouse [84,85,86]. This model also shows a high density of cells with an OS-associated senescent phenotype in different areas of the hippocampus (CA1, CA3, GCL, and SGZ), cortex, and medial septum [82,87,88].

In addition, mitochondrial structural and metabolic dysfunction and chronic oxidative damage have also been found in different cell types including neurons of other DS mouse models such as the Ts16, the Dp16, the Tc1, the Ts1Cje, and the Ts2Cje [89,90,91,92,93,94]. The Tc1 and Ts1Cje DS mouse models do not have triplicated *Sod1* or *App* genes, which as mentioned above, demonstrates that other triplicated genes also play a role in the increased OS and mitochondrial anomalies observed in DS [92,93,94]. Additionally, some of these models also present OS-associated dysfunctions in autophagy and the proteasome degradative systems, which leads to an accumulation of oxidized proteins, alters the proteostasis network, and contributes to neurodegeneration and to cognitive dysfunction [95,96,97,98].

Thus, studies in murine models and DS individuals suggest that an increase in OS and mitochondrial dysfunction could play an important role in the development and progression of cognitive decline in DS. Therefore, the administration of drugs or compounds that reduce the oxidative damage could prevent the neurobiological alterations responsible for the cognitive deficits associated with DS.

## 3. Antioxidant Therapy in DS: Preclinical and Clinical Studies

Most of the preclinical studies that have assessed the efficacy of the administration of antioxidants on the cognitive and neuronal dysfunctions associated with DS have been performed on the Ts65Dn mouse.

### 3.1. Antioxidant Therapies during Adulthood

Several preclinical studies performed on different in vitro and in vivo DS models have demonstrated that lowering OS can reduce the neurobiological and cognitive phenotypes characteristic of DS [68,77,79,81,82,90,99] (Table 1). Supplementation of the diet of Ts65Dn mice with antioxidants such as α-tocopherol or vitamin E reduces OS, attenuates cholinergic neuron degeneration, preserves hippocampal morphology, and improves spatial working memory in this murine model of DS [77]. The neurohormone melatonin exerts multiple antioxidant effects, including being a potent ROS scavenger, modulating anti- and pro-oxidant enzymes, and inducing the recovery of molecules damaged by ROS overgeneration [100,101]. The exogenous chronic administration of melatonin to Ts65Dn mice during adult stages improves spatial learning and memory, restores hippocampal long-term potentiation (LTP) and several neuromorphological alterations, and reduces cholinergic degeneration, hippocampal OS, and the density of senescent cells in the hippocampus [81,82,102].

Based on these and other studies, the efficacy of several antioxidant molecules has been assessed in adults with DS (Table 2). In a randomized, double-blind, placebo-controlled trial, the effects of long-term (over a 2-year period) daily antioxidant supplementation containing α-lipoic acid, ascorbic acid, and α-tocopherol, were evaluated in subjects with DS and dementia. This study demonstrated that long-term supplementation with these antioxidants is safe. However, this treatment did not improve cognitive function nor prevent cognitive decline in individuals with DS [103]. It is important to note that dementia is a serious risk among aging individuals with DS and that current evidence suggests that antioxidant administration is highly unlikely to be helpful at this stage. Consistent with this lack of efficacy, a second study of a longer duration in adults older than 50 years with DS who received ≈672 mg of vitamin E orally (twice daily) over 3 years demonstrated that vitamin E did not delay the cognitive decline of older individuals with DS [104]. Thus, although it has been suggested that vitamin E requires long-term supplementation to induce neuroprotection [105], these studies indicate that supplementation with vitamin E over long periods of time to adults with DS does not protect this population from undergoing neurodegeneration, and it does not delay or reduce the characteristic cognitive decline. However, the lack of pro-cognitive effects reported by Sano et al. [104] in their study may be partially due to other causes. They administered a single antioxidant (vitamin E) which slowly penetrates in the brain only via the α-tocopherol transporter [106], whose expression may be altered by ROS [107]. This may not ensure an adequate supply of this nutrient to the brain. It has also been proposed that the triplication of some genes and the persistent OS beginning from the fetal stages may also induce adaptation through the induction of compensatory mechanisms during aging in DS cells [20,108,109], which could be another reason for the lack of pro-cognitive effects after antioxidant therapy in adults with DS.

However, the results of the clinical trials might have been biased by several confounding methodological factors such as the combination of antioxidants used, the dosage of the different antioxidants administered, or the small sample sizes in the aforementioned clinical trials. Thus, further studies designed to avoid these confounding factors need to be performed to confirm the lack of efficacy of antioxidant supplementation on the cognitive status of DS adults.

The discrepancy found between the efficacy after the administration of vitamin E to the Ts65Dn mouse and to individuals with DS might be due to the differences in the set of triplicated genes between individuals with trisomy 21 and mouse models of DS. Although the Ts65Dn mouse model shares many neuroanatomical, behavioral, and neurobiological similarities, as well as age-related DS phenotypes, including the brain OS profile, with individuals with DS, a limitation of this model is that some of the orthologous genes found in Hsa21 are not triplicated in this model, and other genes that are not triplicated in individuals with DS are in trisomy in the Ts65Dn mouse. Thus, it is possible that the overexpression of several other Hsa21 genes is required to develop the full spectrum of the OS profile observed in individuals with DS and, thus, in this scenario vitamin E may not be as effective as in animal models.

Despite the discrepancies between Hsa21 genes and its murine orthologous genes overexpressed in people with DS and mouse models of this syndrome, because melatonin treatment in adultTs65Dn mice demonstrated antioxidant neuroprotective properties associated with improved cognition, melatonin could serve as a potential therapeutic agent for age-related neurodegeneration and cognitive decline in adults with DS. Due to a variety of physiological and metabolic advantages, the protective effects of this indoleamine against oxidative damage are more potent than those induced by vitamins C or E [100,136]. Furthermore, this indoleamine has been approved for human use, it is normally well tolerated in adults, it does not cause significant adverse events and it is already used in the treatment of other neurodegenerative diseases in which OS is enhanced. Therefore, future clinical trials should assess the efficacy of melatonin to reduce OS, to restore neuronal function and to delay the age-related progression of cognitive alterations in the DS population.

Due to the lack of success in clinical trials after the use of vitamins and conventional antioxidants in adults with DS, preclinical studies are investigating new therapeutic approaches to prevent OS in order to improve cognitive function. The efficacy of other molecules which, instead of acting as ROS scavengers, exert their effects on the molecular pathways implicated in preventing the excessive production of ROS or in the elimination of oxidized proteins, has recently been assessed in adult Ts65Dn mice. In this regard, the administration of a cleavage product of the natural peptide glucagon-like peptide 1 (GLP-1) to adult Ts65Dn animals reduces mitochondrial ROS generation, decreases their alterations in dendritic spine morphology, and improves their electrophysiological (LTP) and cognitive alterations [118]. Associated with increased OS, in DS, disturbances of the mTOR signaling pathway also alters the autophagy system which plays a key role in the cellular response to OS [19]. Additionally, alterations of the brain’s insulin resistance pathway have been associated with the development of AD in DS [137]. The administration of the mTOR pathway inhibitor, Rapamycin, which has been demonstrated to normalize the activity of this pathway and to have antioxidant properties [98], reduced the accumulation of lipoxidized proteins in both the hippocampi and frontal cortices of adult Ts65Dn mice. This treatment also reduced several pathological hallmarks of AD (levels of APP, Aβ-peptides, and hyperphosphorylated tau) and improved the cognitive performance of these animals [98]. Besides these neuroprotective actions, in the Ts65Dn mouse, this inhibition of mTOR signaling after rapamycin administration also prevented the abnormal autophagy and recovered the insulin resistance pathway, thereby decreasing brain insulin resistance [137,138]. Both improvements decrease the risk of developing AD in DS [98,137,138]. Future clinical studies should also assess the efficacy of these compounds at inducing neuroprotection and improving cognition in the DS population through the reduction of ROS production or by the reduction of oxidative damage.

Finally, in DS adults, neurodegeneration is not only driven by increased OS, but as mentioned above, also by other mechanisms such as the development of AD neuropathology (formation of senile plaques or NFTs and accumulation of Aβ peptides) or increased neuroinflammation. Thus, it is possible that combining antioxidants with other molecules that exert neuroprotection through other mechanisms may be more effective at preserving brain health and cognitive abilities in DS adults.

### 3.2. Antioxidant Therapies in Early Life Stages

As mentioned above, OS is present in the brains of individuals with DS from early developmental stages, producing structural and functional neuronal damage that might be irreversible during later life stages. Thus, supplementation with antioxidants should be initiated before persistent OS occurs.

Despite the beneficial effects found in preclinical studies when antioxidant therapy is administered to the Ts65Dn mouse during adult stages, conflicting results have been reported regarding its use at earlier pre- or postnatal stages (i.e., during gestation and/or infancy) (Table 1). Pelsman et al. [99] demonstrated that the administration of SGS-111, an analog of the nootropic Piracetam, to DS human fetal cortical neurons prevented oxidative damage and apoptosis, indicating that this treatment could be effective in rescuing the cognitive alterations that characterize DS. However, in a later study, the pre- and postnatal administration of this compound to Ts65Dn mice did not modify the cognitive abilities of these animals [110]. Similarly, despite melatonin-exerted neuroprotection and induced pro-cognitive effects observed when it was administered during adult stages to Ts65Dn mice, its administration during pre- and postnatal stages did not prevent or reduce the cognitive impairment of these animals. The differences found between the administration of this indoleamine to Ts65Dn mice during early and late stages were probably due to the different doses used in each case, or to the well-known differences in pharmacokinetics between infancy and adulthood [80].

Other compounds with antioxidant properties have been shown to be beneficial in mouse models of DS when administered during early life stages. The flavonoid 7,8-dihydroxyflavone (7,8-DHF) possesses powerful antioxidant properties independent of its actions on the TrkB receptor [139]. Administration of 7,8-DHF to Ts65Dn mice in the postnatal period (from P3 to P15) restores their hippocampal neurogenesis and dendritic spine development, but its effects on the brain are ephemeral since 1 month after cessation of the treatment these mice did not show any learning and memory improvement [111,112]. When this flavonoid was administered from P3 to adolescence (P45–50), it enhanced the learning and memory abilities of Ts65Dn mice; however, no benefits on the cognitive abilities of these animals were found when they received 7,8-DHF during adulthood (from 5 months of age for a period of 6 weeks) [111,112]. Similar to what was previously mentioned regarding the effects of melatonin administration, these results indicate that the timing of the administration of some antioxidants is critical for the attainment of positive effects on the brain. However, this might not be the case for all antioxidants, treatment with vitamin E during early life stages ameliorated oxidative stress and, similar to when it was administered during adulthood, it also improved cognition in the Ts65Dn mouse [79]. Finally, preliminary studies have shown that apigenin, an FDA-approved antioxidant small molecule, when administered to pregnant female mice, significantly improved the postnatal exploratory behavior in the open field of their Ts1Cje pups, and thus it may serve as a potential candidate for prenatal therapy [113,140]. Future studies should explore its effects in later life-stages.

Early supplementation of antioxidants to children and teenagers with DS, such as vitamins E and C, attenuates systemic oxidative damage [38,131,132] (Table 2). In a randomized controlled trial, Mustafa et al. [132] evaluated the ability of the administration of two antioxidants, vitamin E (266 mg/day) and α-lipoic acid (100 mg/day) over a 4-month period, to reduce lipid peroxidation and DNA damage in serum and urine, respectively, in 93 children and teenagers ranging from 7 to 15 years of age. While none of the antioxidants reduced the lipid peroxidation markers in serum, α-tocopherol slightly decreased oxidative stress at the DNA level in DS children. In another study, daily antioxidant treatment with a combination of vitamins E (400 mg/day) and C (500 mg/day) given to children and teenagers with DS over a 6-month period decreased lipid peroxidation in the blood of DS subjects [131]. In a later study, the same group also demonstrated that administration of this combined antioxidant therapy persistently attenuated the systemic oxidative damage even when they assessed it 6 months after the discontinuation of the treatment [38]. Therefore, longer treatments or higher doses such as the ones used in the later studies may be necessary to reduce oxidative damage to lipids with vitamin E supplementation in individuals with DS.

Other nutrients with antioxidant properties such as minerals, fatty acids, or amino acids have also been tested in individuals with DS (Table 2). Despite promising results after prolonged Coenzyme Q10 treatment (4 mg/kg/day for 20 months) administered to children and teenagers with DS where Tiano et al. [65] found a reduction in DNA damage in peripheral blood leukocytes, long-term supplementation (4 years) of this coenzyme did not affect RNA or DNA oxidation in children with DS [133]. The different results reported between these studies may be due to the different biomarkers used to quantify oxidative stress and antioxidant activity, or to the different cells or tissues used for each study. Additionally, considerable attention has been given to selenium as an essential micronutrient with important antioxidant actions through the modulation of the activity of antioxidant enzymes such as GPx [141]. However, its supplementation over a period of 6 months to children with DS ranging from 6 months to 16 years of age caused effects opposite to those expected, since this treatment reduced the activity of this antioxidant enzyme in erythrocytes [134]. In that study, selenium supplementation was well tolerated, and no side effects were observed, but the authors do not recommend its supplementation to individuals with DS. In contrast, dietary supplementation for several treatment cycles (one treatment cycle = 30 days dietary supplementation followed by a 30-day wash-out period) with the antioxidants α-lipoic acid and L-cysteine, which act by modulating the glutathione system, decreased serum ROS in children with DS [135].

Although the aforementioned studies suggest that supplementation with antioxidants (mainly antioxidant vitamins) may alleviate OS in children and young adults with DS, clinical trials have shown that minerals, vitamins, or antioxidant supplementation provides little or no benefit to the cognitive function of children with DS [120,121,122,123,124] (Table 2). Ellis et al. [125] demonstrated that oral supplementation with a daily combination of antioxidants (selenium, zinc, vitamins A, E, and C) and folinic acid to infants less than 7 months old with trisomy 21 for 3–8 months did not produce any effect on language acquisition, psychomotor development, or in the levels of certain biochemical markers of OS. These authors propose that the lack of beneficial effects could be due to the low dose of the supplements or the short duration of the treatment (18 months) [125]. In agreement with this idea, in a later study, oral daily long-term (12 months) folate supplementation at high doses to children with DS (aged between 3 and 30 months) slightly improved their psychomotor development but did not produce any changes in sociability, language, or coordination. The authors attribute the lack of benefits found in this study after the antioxidant treatment to the wide range of ages of the participants, which caused heterogeneity in the results of some of the variables assessed [126]. However, in another study, after 6 months of supplementation with a mixture of different nutrients with antioxidant properties including minerals (zinc) and vitamins (vitamins A, C, E, B1, B2, B3, B6, B9, B12) to older children with ages ranging between 5 and 16 years, cholinesterase activity increased in the serum of children with DS. This nutritional supplementation also produced a significant improvement in their cognitive skills and behavioral patterns [127].

Altogether, the outcome of these studies suggests that future clinical trials to evaluate the effect of antioxidant supplementation at early life stages should employ higher doses over longer periods of time, a larger number of subjects, a narrower age range, and should include programs to periodically monitor whether earlier intervention with antioxidants in DS exerts beneficial effects in cognition, and if so, whether these effects are maintained over time. Finally, because OS in DS begins during fetal stages, in subsequent clinical studies, antioxidant therapies should begin during pregnancy after diagnosis of trisomy 21, or in the early stages of postnatal life.

## 4. Targeting Mitochondrial Dysfunction to Reduce OS in DS

Mitochondria are the main source of ROS production, but are also one of the main targets of ROS, therefore, new therapeutic approaches to promote mitochondrial function and/or reduce mitochondrial damage and energy deficits in DS have emerged (Table 1 and Table 2).

Several preclinical and clinical studies have demonstrated that plants and plant extracts protect against neurodevelopmental and neurodegenerative diseases. They exert neuroprotection by diverse mechanisms depending on their ability to act in various signaling pathways, including those related to OS [142,143,144].

Acetyl-l-carnitine is a molecule derived from the acetylation of carnitine that is naturally produced by the body, but it can also be taken as a dietary supplement. This molecule has been demonstrated to possess multiple antioxidant properties in the nervous system, most of them related to its ability to act on mitochondrial metabolism [145,146]. In a double-blind study that enrolled 40 adults with DS between the ages of 18 and 30 years, acetyl-l-carnitine was administered in ascending doses: 10 mg/kg/day in the first month, 20 mg/kg/day in the second month, and 30 mg/kg/day for the rest of the study. The individuals received the treatment for a 6-month period, followed by a 3-month wash-out period. Psychological and behavioral assessments were performed at the start of the study and at 3, 6, and 9 months after the start of the treatment. The authors report that the cognitive abilities, behavioral problems, and daily living skills of the participants who received acetyl-l-carnitine did not differ from those who received placebo at any time point of the assessments [128].

Polyphenols are secondary metabolites produced by plants and are important constituents of the human diet. Plant polyphenol compounds produce potent neuroprotective effects because of their ability to act over a variety of signaling proteins that affect mitochondrial homeostasis, decrease OS, reduce Aβ accumulation, reduce neuroinflammation, and decrease cognitive decline in various neurodevelopmental and neurodegenerative diseases, including DS [142,143,144]. To the best of our knowledge, the only polyphenols that have been investigated as potential therapeutic tools in DS are epigallocatechin-3-gallate (EGCG), resveratrol, hydroxytyrosol (HT), and curcumin.

Resveratrol, a polyphenol isolated from grapes, red wine, peanuts, and berries, and EGCG, the most common bioactive catechin present in green tea, promote neuroprotection by different mechanisms, including their antioxidant actions and their effects on molecular pathways implicated in the maintenance of mitochondrial homeostasis [33]. Furthermore, EGCG is a potent and selective inhibitor of DYRK1A activity, a kinase encoded by the Hsa21 gene *DYRK1A*, and thus overexpressed in DS, that participates in numerous molecular pathways related to some of the altered phenotypes of this syndrome. Administration of these polyphenols restores the impairment of mitochondrial bioenergetics and biogenesis, improving the activity of mitochondrial respiratory chain complexes and ATP production, and promotes the proliferation of neuronal progenitor cells isolated from the hippocampus of the Ts65Dn mouse [83]. In vivo, oral administration of EGCG to Ts65Dn mice (2–3 mg/day) normalizes the activity of DYRK1A and improves the cognitive abilities of these mice [114]. In addition, these authors also reported that oral administration of EGCG to adults with DS (9 mg/kg/day) for 6 months also produced benefits in different cognitive parameters [116]. However, the effects of EGCG when administered to Ts65Dn mice at early life stages are controversial. Stagni et al. [114] administered EGCG during prenatal stages to Ts65Dn mice. Although this compound exerted beneficial effects on neurogenesis immediately after the discontinuation of the treatment, these effects were not maintained 1 month later [114]. In addition, EGCG administered at early postnatal stages to Ts65Dn mice failed to improve cognition and induced skeletal anomalies [115]. Thus, similar to the aforementioned results obtained after the administration of different antioxidants in early stages in DS, studies with EGCG in children with this syndrome must carefully choose the dosage and duration of the treatment at different life stages in order to obtain beneficial effects on cognition while avoiding undesirable side-effects.

The safety and efficacy of EGCG has also been tested in various clinical trials with people with DS. A randomized, double-blind, placebo-controlled, phase 2 trial conducted with young adults with DS treated with EGCG (9 mg/kg/day) for 12 months improved visual recognition memory, inhibitory control, and adaptive behavior [129]. In a case report, Vacca and Valenti, [130] evaluated the beneficial effects of a single daily dose of a supplement that combined EGCG (10 mg/kg/day) plus omega-3 fish oil (8 mg/kg/day) given to a DS child over a 6-month period. The results showed that the combination of these compounds restored mitochondrial deficits and improved neuropsychological performance without producing side effects.

In the case of resveratrol, its effectiveness has been questioned because of its low bioavailability, and so far its efficacy on the DS population has not been assessed.

HT, a polyphenol found in olives and olive oil, has been shown to display ROS scavenging and chelating properties by decreasing ROS generation and lipid peroxidation and reducing intracellular iron accumulation, respectively, in DS erythrocytes [147].

Curcumin is another natural polyphenol commonly used as a food additive. Recently, we tested its ability as a neuroprotective compound to rescue the neuromorphological and cognitive alterations of the Ts65Dn mouse when it is administered prenatally or during early postnatal stages. In that study, prenatal administration of curcumin increased the brain weight as well as the density of proliferating and mature hippocampal cells, and produced a long-term improvement of cognition in the Ts65Dn mouse, while its postnatal administration did not induce any beneficial effect in the altered phenotypes of these animals [117]. However, the effects of curcumin administration on OS or mitochondrial function were not evaluated in this model of DS. As curcumin also exerts neuroprotection due to its antioxidant and anti-inflammatory actions, future studies should address whether the curcumin-induced pro-cognitive effect in this model is mediated by its effects on redox and mitochondrial status. Additionally, as curcumin is currently being evaluated in clinical trials [148,149], its antioxidant and pro-cognitive effects should also be assessed in the DS population.

Finally, it has been demonstrated that the administration of compounds that act on specific mitochondrial regulatory genes could be a promising therapy to improve mitochondrial dysfunction in DS. One of them is metformin, an FDA-approved drug that has been shown to rescue mitochondrial dysfunction by increasing the respiratory activity and the cellular ATP content, and also to reduce the anomalies found in mitochondrial morphology in fibroblasts from DS fetuses [24]. It has also been demonstrated that Pioglitazone, another FDA-approved drug for the treatment of diabetes, improves mitochondrial bioenergetics by increasing the ATP content and the oxygen consumption rate, and by decreasing ROS production in trisomic fetal fibroblasts [119]. However, to date, no clinical trials have been performed to assess its efficacy in humans with DS.

These recent studies indicate that the administration of natural polyphenols or other mitochondrial therapies may improve the oxidative status, energy metabolism and cellular function in DS individuals, avoiding premature neurodegeneration and delaying the progression of the cognitive deficits associated with this syndrome.

## 5. Conclusions

In the last few decades, the life expectancy of individuals with DS has increased considerably and today is above the 60 s [150]. Although various pharmacotherapies have been proposed [140,151,152,153], there is currently no effective treatment available to improve the cognitive disabilities of individuals with DS. As there is compelling evidence that OS is one of the main mechanisms implicated in the neurodevelopmental anomalies and the neurodegeneration that occurs during aging in this condition, many preclinical and clinical studies using antioxidants to decrease oxidative damage and improve cognition in DS have been conducted.

However, the benefits observed in preclinical studies after antioxidant therapies on the cognitive abilities of murine models of DS stand in contrast to those obtained in clinical trials. Most available clinical studies demonstrate that antioxidant supplementation reduces biomarkers of oxidative stress and promotes antioxidant activity in people with DS, but failed to find either an improvement in cognitive functioning or a stabilization of the cognitive decline in aged DS individuals. Therefore, the beneficial effects of antioxidant therapies on the cognitive dysfunction in individuals with DS is still a matter of debate and, at present, the reasons for the relative failure of antioxidant interventions in DS are not known.

Although in vitro and in vivo DS models are useful in order to study the molecular mechanisms implicated in brain phenotypes, as well as for screening new therapeutic molecules prior to initiating clinical trials, they do not fully recapitulate the complex behavioral and cognitive phenotypes and the genotype of the DS population [153], demonstrating that there is a ‘biological gap’ between studies performed in cellular and mouse DS models and those conducted with individuals with DS. This fact makes it difficult to translate the results obtained in DS models after antioxidant therapy into safe and effective treatments for human subjects.

Moreover, several other factors may also contribute to failures in the translation of these antioxidant therapies to the DS population. Among these factors are the different drug formulations, interspecies differences in drug pharmacokinetic characteristics, the adequacy of DS models, the timing of treatments, the lack of human translational endpoints and of standardized outcome measures, and the low number of participants in clinical trials, among others [153].

As the results of trials using either a single antioxidant (vitamin E) or a combination of antioxidants have not provided a unifying outcome, further investigation into antioxidant therapy for the treatment of cognitive disturbances in DS is needed. In this regard, several recently conducted mitochondria-targeted antioxidant interventions have led to promising results and they have therefore been proposed to be useful as antioxidant and antiaging therapies in DS. Among them are several natural polyphenols and other mitochondrial therapies that should be evaluated in future clinical trials. In addition, some of these molecules are being used in humans. Another possibility when planning future trials with people with DS could be the election of a combination of different antioxidant therapies that play a role in both mitochondrial dysfunction and counteracting oxidative damage, to reduce OS and improve cognition.

Finally, due to the strong connection between OS and other neuropathological mechanisms that also contributes to neurodegenerative processes in DS, future therapeutic approaches may also combine cellular antioxidants with molecules that act on other altered process such as the prevention of Aβ aggregation, the reduction of neuroinflammatory processes, the modulation of energy metabolism, the regulation of protein degradative systems (autophagy and proteasome), exosome cargo and secretion, and the acquisition of cellular senescent phenotypes, among others.

## Figures and Tables

**Table 1 antioxidants-09-00692-t001:** Preclinical studies that assessed the efficacy of natural and synthetic molecules with antioxidant properties in cellular and murine models of Down syndrome (DS).

Antioxidant Drug	Model of DS	Dosage and Treatment Duration	Results	References
Vitamin E (α-tocopherol)	Adult Ts65Dn mice (4 months of age)	50 ± 5 mg/kg/day supplemented in the diet for 5 months	Improvement of spatial memory, reduction of cholinergic neurodegeneration, normalization of OS markers	[77]
	Pregnant Ts65Dn females and their pups	Pregnant Ts65Dn females received (0.1% (*w*/*w*) α-tocopherol acetate per kilogram of diet) from the day of conception throughout the pregnancy and the pups received the same diet from the day of birth for 12 weeks	Reduction of levels of lipid peroxidation products, attenuation of cognitive impairment, improvement of the hippocampal hypocellularity	[79]
SGS-111	DS cortical neurons cultures	Various doses from 10 nM to 100 µM (30 min before the addition of H_2_O_2_ to the cultures and until 24 h later)	Inhibition of the accumulation of intracellular free radicals and lipid peroxidation damage in neurons treated with H_2_O_2_. Reduction of the appearance of degenerative changes and increment of neuronal survival	[99]
	Adult Ts65Dn mice (6 months of age). Pregnant Ts65Dn females and their pups over their entire life (5 months)	In both studies: 0.5 mg/kg (daily subcutaneously injected). Duration of adult treatment: 6 weeks. Duration of pre- and postnatal treatment: 5 months	No evidence of changes in behavior or cognition	[110]
Melatonin	Adult Ts65Dn (6 months of age)	0.5 mg/day in their drinking water for 6 months	Improvement in spatial learning, reduction of cholinergic neurodegeneration, improvement of hippocampal neurogenesis, reduction of synaptic inhibition, restoration of hippocampal LTP, reduction of protein and lipid oxidative damage and of the density of senescent cells in the hippocampus	[81,82,102]
	Pregnant Ts65Dn females and their pups	0.5 mg/day in their drinking water during pregnancy to TS females until the weaning of the offspring, and to the pups until the age of 5 months	No effect on cognitive or neurogenesis abnormalities. Modulation of antioxidant enzymes: SOD in the cortex, and catalase in the hippocampus. No effect on lipid and protein oxidative damage	[80]
7, 8-dihydroxyflavone	Ts65Dn pups, young and adult stages	In all studies: 5 mg/kg (daily subcutaneously injected). Postnatal treatment: for 12 days. Adolescent treatment: from P3 to adolescence (P45–50) Adult treatment: 6 weeks	Postnatal treatment: restoration of hippocampal neurogenesis and dendritic spine development, but 1 month after cessation of the treatment there was no evidence of pro-cognitive effects. Adolescent treatment: improvement in cognition. Adult treatment: no effect on cognition	[111,112]
Apigenin	Ts1Cje mothers and their pups	200–250 mg/kg/day in chow, during pregnancy to the mothers and to their pups up until 8–10 weeks of postnatal life	Improvement of exploratory behavior	[113]
Epigallocatechin-3-gallate (EGCG)and Resveratrol	Neuronal progenitor cells isolated from the hippocampus of the Ts65Dn mouse	EGCG and Resveratrol, 20 μM and 10 μM, respectively, for 24 h	Restoration of mitochondrial homeostasis and promotion of proliferation in neuronal progenitors	[83]
Epigallocatechin-3-gallate	Ts65Dn pups	25 mg/kg in a daily subcutaneous injection from postnatal day 3 to postnatal day 15	At P15 the treatment rescues hippocampal neurogenesis. This effect was not evident 1 month later after cessation of the treatment	[114]
	Ts65Dn pups	0.4 mg/mL in their drinking water (≈50 mg/kg/day) from postnatal day 24 to postnatal day 68 (≈6 weeks)	No improvement in cognitive deficits and produced detrimental skeletal effects	[115]
	Young adult Ts65Dn (3 months of age)	90 mg/mL for a dose of 2–3 mg per day in drinking water for 1 month	Improvement of hippocampal-dependent learning deficits	[116]
Curcumin	Pregnant Ts65Dn mice and their pups and young mice	In both studies: 300 mg/kg in a daily subcutaneous injection. Prenatal treatment: from embryonic day 10 to postnatal day 2. Postnatal treatment: from postnatal day 2 to postnatal day 15	Prenatal effects: increase in brain weight, cell proliferation, and pro-cognitive long-term effects. Postnatal effects: no effect on cognition	[117]
Glucagon-like peptide 1	Adult Ts65Dn mice (9 months of age)	500 ng/g daily via intraperitoneal injection for 2–3 weeks	Reduction of mitochondrial ROS generation, of dendritic spine morphology alterations, and improvement of LTP and cognitive alterations	[118]
Rapamycin	Adult Ts65Dn mice (6 months of age)	Three times per week with a dose of 0.1 μg/μL (1 μg/mouse) by intranasal route for 12 weeks	Restoration of mTOR pathway and reduction of lipoxidized proteins, rescue of autophagy and insulin signaling. Improvement in cognition	[96,98]
Pioglitazone	Trisomic fetal fibroblasts	5 mM for 3 days	Improvement of mitochondrial bioenergetics: increase of basal ATP content and oxygen consumption rate and decrease of ROS production	[119]
Metformin	Trisomic fetal fibroblasts	0.05 or 0.5 mM for 3 days	Reduction of mitochondrial abnormalities: increases in ATP production, oxygen consumption rate, and mitochondrial activity. Reversion of mitochondrial fragmentation and promotion of mitochondrial network	[24]

**Table 2 antioxidants-09-00692-t002:** Clinical trials that assessed the efficacy of natural and synthetic molecules with antioxidant properties in DS subjects.

Type of Trial	Antioxidant	Subjects/Cell Type	Dosage and Treatment Duration	Results	References
R, DB, PC	α-tocopherol, ascorbic acid and α-lipoic acid	53 individuals with DS and dementia (average age ≈ 50 years)	Daily dose of 900 IU of α-tocopherol, 200 mg of ascorbic acid, and 600 mg of α-lipoic acid for 2 years	No improvement in cognitive functioning or stabilization of cognitive decline	[103]
R, DB, PC	Vitamin E (α-tocopherol)	337 adults with DS older than 50 years of age	1000 IU of vitamin E, twice daily for 3 years	No retardation in the progression of cognitive deterioration	[104]
DB, case-control study, PC crossover trial	Mixture of vitamins and minerals	A total of 115 children with DS (in the four studies) aged between 7.5 months and 17 years old	Between 4 and 8 months depending on the study	No significant effect on development or behavior. No effect on intelligence quotient tests. No effect on standard psychological tests. Induction of various side-effects	[120,121,122,123,124]
R, PC	Mixture of vitamins and minerals	156 children with DS, less than 7 months old	Daily supplementation with 10 μg of selenium, 5 mg of zinc, 0.9 mg of vitamin A, 100 mg of vitamin E, 50 mg of vitamin C, and 0.1 mg of folinic acid for 18 months	No benefits in psychomotor development, language acquisition or in the levels of markers of OS in red blood cells	[125]
R, DB	Leucovorin (folinic acid)	117 children with DS aged between 3 and 30 months old	Daily dose of 1 ± 0.3 mg/kg for 12 months	Improvement of psychomotor development. No effect on sociability, language or coordination	[126]
Pilot study	Mixture of nutrients zinc, vitamins (A, C, E, B1, B2, B3, B6, B9, B12) and minerals	40 children with DS aged between 5 and 16 years old	5000 IU of vitamin A, 25 IU of vitamin E, 100 mg of ascorbic acid, 10 mg of thiamine mononitrate, 10 mg of riboflavin, 3 mg of pyridoxine hydrochloride, 5 µg of cyanocobalamin, 50 mg of niacinamide, 1 mg of folic acid, 12.5 mg of calcium pantothenate, 2.5 mg of copper, 60 µg of selenium, 1.4 mg of manganese and 5 µg of chromium for 6 months	Reduction of serum acetyl- and Butyrylcholinesterase. Improvement in cognitive skills and behavioral patterns	[127]
DB	Acetyl-l-carnitine	40 adults with DS aged between 18 and 30 years old	Ascending doses: 10 mg/kg/day for the first month, 20 mg/kg/day for the second month and afterwards 30 mg/kg/day for the rest of the study. Duration: 6 months, followed by a 3-month “wash-out” period	No effect on cognitive abilities, behavioral problems or daily living skills	[128]
Pilot study	Epigallocatechin-3-gallate (EGCG)	31 young adults with DS aged between 14 and 29 years old	9 mg/kg/day for 6 months	Positive effects on memory recognition, working memory, and quality of life	[116]
R, DB, PC	EGCG	84 young adults with DS aged between 16 and 34 years old	9 mg/kg/day for 12 months	Improvement in visual recognition memory, inhibitory control, and adaptive behavior	[129]
CR	EGCG plus omega-3 fish oil	One DS child, 10 years and 3 months old	EGCG: 10 mg/kg/day and omega-3 fish oil: 8 mg/kg/day for 6 months	Improvement in the ability to perform tasks requiring concentration. Restoration of mitochondrial respiratory chain complex activities in lymphocytes from peripheral blood	[130]
	Vitamin E, vitamin C	Healthy and DS children between 3 and 14 years of age	Vitamin C (500 mg/day), vitamin E (400 mg/day) administered daily for 6 months	Attenuation of systemic oxidative damage in the blood of DS subjects. These effects persisted for at least 6 months after the cessation of the antioxidant therapy	[38,131]
	α-tocopherol or α-lipoic acid	93 DS children between 7 and 15 years of age	α-tocopherol (400 IU/day) or α-lipoic acid (100 mg/day) for 4 months	Attenuation of OS at the DNA level in serum after 20 months of treatment. No effect on RNA or DNA oxidation after 4 years of treatment	[132]
	Coenzyme Q10	Children (aged 5–12 years, *n* = 20) and teenagers (aged 13–17 years, *n* = 8) with DS	4 mg/kg/day for 20 months (children), and 4 years (teenagers)	Reduced the activity of GPx in erythrocytes	[65,133]
	Selenium	48 children with DS (aged between 6 months and 16 years)	10 µg/kg body weight/day for 6 months	Improvement of the rest-activity rhythms	[134]
	α-lipoic acid and L-cysteine	20 children with DS (aged between 1 and 16 years old) with redox imbalance	One capsule per day that contained 200 mg of α-lipoic acid, and 200 mg of L-cysteine, over several treatment cycles (one treatment cycle = 30 days plus a 30 day wash-out period)	Reduction of serum ROS	[135]

R = randomized; DB = double-blind; PC = placebo-controlled; CR = case report.

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
