# Peer review of "Antioxidants in Down Syndrome: From Preclinical Studies to Clinical Trials"

_antioxidants, 2020, doi:10.3390/antiox9080692_

Round 1
Reviewer 1 Report
In this review, the authors describe the potential role of antioxidants in Down Syndrome (DS); despite this interesting field, the manuscript doesn’t analyze the whole spectrum of antioxidants drugs.
To improve the manuscript I suggest:
- Add a new section in which mechanisms of oxidative stress is involved in DS (SOD is not the unique mechanism), If possible I suggest to insert an image;
- Increase the synthetic and natural antioxidants drugs analyzed;
- Create a table whit name of antioxidants drugs, dose, time used and bibliography;
- I suggest to read Di Carlo et al., Free Radical Research 2012 because is an interesting paper and the author could have some suggestion.
Minor revision
Page 8 line 391, remove 10;
Revises the formatting of the bibliography.
Reviewer 2 Report
Down syndrome (DS) is one of the most common genetic causes of cognitive dysfunction. However, there is no effective pharmacological therapy to improve cognitive deficits in DS individuals.
Once oxidative stress is one of the most relevant neuropathological mechanisms in these patients, antioxidants might be good candidates as neuroprotective agents in order to improve their cognitive function.
This manuscript is original, well written and organized, and within the scope of Antioxidants journal. In this review, authors summarize and discuss several preclinical and clinical studies in which the effect of some isolated antioxidants (or the combination of few antioxidants) is evaluated. However, considering the increasing importance of medicinal plants and dietary supplements industry, authors must include studies on the impact of the use of antioxidant botanicals (composed by complex mixtures of phytochemicals), such as green and white teas, in the prevention and / or treatment of oxidative stress and other neurophatological processes that occur in DS individuals.
Reviewer 3 Report
In this manuscript Drs. Rueda Revilla and Martínez-Cué reviewed contribution of oxidative stress to the CNS pathology associated with Down syndrome (DS) and possible role of anti-oxidants as a therapy aim to ameliorate CNS dysfunction. In DS, overexpression of several chromosome 21 genes, effects cellular and systemic oxidative stress, which is one of the most important neuropathological processes that contributes to the cognitive deficits and multiple neuronal alterations in DS. Thus, therapeutic interventions that reduce OS have been proposed as a promising strategy to avoid neurodegeneration and potentially to improve cognition in DS patients. This timely review summarizes preclinical studies of cell cultures and mouse models, and clinical studies of anti-oxidant therapies which also reduce mitochondrial alterations on the cognitive dysfunction associated with DS have been assessed. This is a robust, well-written review based on currently available literature.
Author Response
Thanks for your comments.
Reviewer 4 Report
In this paper, the authors review the use of antioxidants in persons with Down syndrome (DS) and mouse models thereof. Generally, this review is accurate, if repetitive of what has already been published. This reviewer only has a few suggestions to strengthen the paper.
- While the authors do a good job in indicating caveats to conclusions or lack thereof of studies discussed, some additional inclusions could augment their points: a) On Line 157ff, authors might want to point out that by age 50, most DS individuals typically are essentially demented. That is, neurons are already dead. Antioxidant administration at age 50 is highly unlikely to be helpful. b) Lines 168-172, regarding reference 87, the antioxidant employed vitamin E, which only slowly penetrates the brain via a tocopherol transporter.
- The study in reference (81) demonstrated that rapamycin use in the Ts65Dn mouse not only significantly decreased oxidative stress in brain, but this observation was correlated to decreased AD neuropathology and increased cognitive performance. Moreover, rapamycin inhibits mTOR (as the authors note), but they could add that such inhibition leads to restoration of autophagy (which itself is inhibited by activation of mTORC1) and decreased insulin resistance, both expected to decrease risk of transition to AD in DS individuals. These notions should be added to this part of the review.
- The authors briefly mention the gene on Chr 21, Bach1, which is a transcriptional repressor of heme oxygenase-1 (HO-1). The protein HO-1 is related to decreased oxidative stress by scavenging released heme, a pro-oxidant moiety, producing biliverdin. The latter, in turn, is transformed to bilirubin by biliverdin reductase-A (BVR-A), and at low levels bilirubin is a good antioxidant. This could be added to the MS.
- Minor: On page 3, Lines 98,99: several key references are cited by name instead of reference numbers.
Round 2
Reviewer 1 Report
I would like to thank the authors for their excellent manuscript revision.
Reviewer 2 Report
Based on reviewers comments and suggestions, authors considerably improved their manuscript.